# Characterization of Novel Rhabdoviruses in Chinese Bats

**DOI:** 10.3390/v13010064

**Published:** 2021-01-05

**Authors:** Dong-Sheng Luo, Bei Li, Xu-Rui Shen, Ren-Di Jiang, Yan Zhu, Jia Wu, Yi Fan, Hervé Bourhy, Ben Hu, Xing-Yi Ge, Zheng-Li Shi, Laurent Dacheux

**Affiliations:** 1CAS Key Laboratory of Special Pathogens and Biosafety, Wuhan Institute of Virology, Chinese Academy of Sciences, Wuhan 430071, China; dongshengluo@outlook.com (D.-S.L.); libei@wh.iov.cn (B.L.); pukulovesu@163.com (X.-R.S.); drteamwind1004@163.com (R.-D.J.); zhuyan@wh.iov.cn (Y.Z.); wuj@wh.iov.cn (J.W.); yifanfs0224@163.com (Y.F.); huben@wh.iov.cn (B.H.); 2University of Chinese Academy of Sciences, Beijing 100049, China; 3Institut Pasteur, Lyssavirus Epidemiology and Neuropathology Unit, 75724 Paris, France; herve.bourhy@pasteur.fr; 4Hunan Provincial Key Laboratory of Medical Virology, College of Biology, Hunan University, Changsha 410082, China; xyge@hnu.edu.cn

**Keywords:** bat, rhabdovirus, lyssavirus, vesiculovirus, ledantevirus, China, RT-qPCR, phylogeny, brain, complete genome

## Abstract

Bats, the second largest order of mammals worldwide, harbor specific characteristics such as sustaining flight, a special immune system, unique habits, and ecological niches. In addition, they are the natural reservoirs of a variety of emerging or re-emerging zoonotic pathogens. *Rhabdoviridae* is one of the most diverse families of RNA viruses, which consists of 20 ecologically diverse genera, infecting plants, mammals, birds, reptiles, and fish. To date, three bat-related genera are described, named *Lyssavirus*, *Vesiculovirus*, and *Ledantevirus*. However, the prevalence and the distribution of these bat-related rhabdoviruses remain largely unknown, especially in China. To fill this gap, we performed a large molecular retrospective study based on the real-time reverse transcription polymerase chain reaction (RT-qPCR) detection of lyssavirus in bat samples (1044 brain and 3532 saliva samples, from 63 different bat species) originating from 21 provinces of China during 2006–2018. None of them were positive for lyssavirus, but six bat brains (0.6%) of *Rhinolophus* bat species, originating from Hubei and Hainan provinces, were positive for vesiculoviruses or ledanteviruses. Based on complete genomes, these viruses were phylogenetically classified into three putative new species, tentatively named Yinshui bat virus (YSBV), Taiyi bat virus (TYBV), and Qiongzhong bat virus (QZBV). These results indicate the novel rhabdoviruses circulated in different Chinese bat populations.

## 1. Introduction

Bats belong to the second largest mammalian order, Chiroptera, which includes over 1350 extant species belonging to 21 bat families, within the two suborders, Yinpterochiroptera and Yangochiroptera [1,2]. These mammals have an extensive geographical distribution and present various unique characteristics such as the capability of sustaining flight and migration, a gregarious behavior, and an important longevity compared with their size and their lifestyle [3]. In addition, multiple lines of evidence indicate that bats are infected with or can host various pathogens, mainly viruses, without displaying clinical symptoms [4,5]. This ability to resist or tolerate viral infection may be directly related to their special immune system [6,7,8]. Over 200 viruses of 27 families were isolated or detected in bats, and numerous viruses, including zoonotic agents, originate from bats [9,10,11]. Indeed, in the past two decades, numerous emerging or re-emerging infectious diseases were demonstrated or suspected to be connected with bats [11], including infections associated with coronavirus with severe acute respiratory syndrome (SARS) [12,13]; Middle-East respiratory syndrome (MERS) [14]; and, more recently, Coronavirus Disease 2019 (COVID-19) [15], as well as acute fatal hemorrhagic diseases caused by filoviruses (Ebola and Marburg viruses) [16,17] and henipaviruses (Hendra and Nipah viruses) [18,19,20], and viral encephalitis with rabies caused by lyssavirus infection [21,22,23,24].

For the latter, 17 currently recognized species and one tentative species have been described so far (https://talk.ictvonline.org/), which are all suspected to induce rabies. Nearly all of them were found in bats, indicating that these animals are the original and natural reservoir of lyssaviruses [25,26,27]. These viruses belong to the genus *Lyssavirus* in the family *Rhabdoviridae*, with *Rabies lyssavirus* (including rabies virus—RABV) acting as the prototype species (https://talk.ictvonline.org/). Apart from lyssaviruses, rhabdoviruses of two other genera have been found in bats, with *Vesiculovirus* and *Ledantevirus*. Within the genus *Ledantevirus*, two bat-associated members have been reported in America [28] and China [29], and several different ledanteviruses species have been previously reported in Africa [30,31] and Europe [32]. As for all rhabdoviruses, these bat-related rhabdoviruses are typically enveloped virions with bullet-shaped or bacilliform morphology, and are reported to be in the range of 100–460 nm in length and 45–100 nm in diameter. They are characterized by a linear single stranded negative-sense RNA genome of approximately 10–16 kb with five canonical genes encoding the nucleoprotein (N), the phosphoprotein (P), the matrix protein (M), the glycoprotein (G), and the RNA-dependent RNA polymerase (L) [33,34]. However, many rhabdovirus genomes also encode multiple additional accessory proteins such as vesiculoviruses and lyssaviruses, which commonly encode small proteins in alternative open reading frames (ORFs) in the P gene [35,36], and abundant accessory ORFs were reported between the P and M gene of hapavirus genomes [33,37].

The family *Rhabdoviridae* exhibits a large ecological diversity with members infecting plants or animals including mammals, birds, reptiles, insects, or fish [33,38,39], and some genera also including many important animal and human viruses such as *Lyssavirus* (e.g., rabies virus) [21,22,24] and *Vesiculovius* (e.g., Chandipura virus) with acute encephalitis in humans [40], *Tibrovirus* (e.g., Bas-Congo virus) with acute hemorrhagic fever in humans [41], or *Vesiculovirus* and *Ephemerovirus* affecting cattle [42,43]. There is increasing evidence that bats are playing an important role in rhabdovirus spillover and highlight the importance of performing active surveillance of this animal reservoir, a source of viral genetic diversity and potential rhabdovirus emergences. Indeed, new members in *Lyssavirus*, *Vesiculovirus*, and *Lendatevirus* genera have recently been found in bats. In addition, the presence of a wide diversity of rhabdoviral sequences has been described in bat samples and in bat ectoparasites, especially in Europe [44,45].

However, studies to investigate the presence of lyssavirus and, to a less extent, bat-associated rhabdoviruses, are limited in Asia, especially in Chinese bats. To date, two lyssaviruses isolated in Jilin and Taiwan have been reported [46,47], and rabies virus seroprevalence studies in bats have been conducted in the South of China [48]. Within the genus *Vesiculovirus*, only one bat-associated virus has been reported in Yunnan in 2017 [29]. No other data are available regarding the circulation of virus members belonging to the genus *Ledantevirus* or to other genera. To further investigate the prevalence of lyssavirus, and more generally rhabdovirus, in bats in China, we conducted a large retrospective study based on molecular detection, based on the analysis of 4576 samples (1044 brain and 3532 saliva samples) from 63 different bat species, covering a period of 13 years (2006–2018). Based on this large screening, we were not able to detect the presence of any lyssavirus. Three putative novel bat rhabdoviruses were identified from *Rhinolophus sinicus* or *Rhinolophus affinis* in two different locations, with Taiyi bat virus (TYBV) belonging to the genus *Ledantevirus*, and with Yinshui bat virus (YSBV) and Qiongzhong bat virus (QZBV), both related to the genus *Vesiculovirus*.

## 2. Materials and Methods

### 2.1. Ethics Statement

Sample collection and animal experiments were performed according to standard procedures approved by the Animal Ethics Committee of the Wuhan Institute of Virology (WIVH05210201 and WIVA05201703).

### 2.2. Sample Collection and Storage

Bat samples (brain and saliva) were part of the biocollection housed in the Wuhan Institute of Virology, Chinese Academy of Sciences. They were originally collected from the field, during the period 2006–2018. Bat brain samples were transported from the field in liquid nitrogen, and saliva swab samples were immersed in 1 mL Virus Transport Medium (VTM, pH 7.4 Hank’s balanced salt solution containing bovine serum albumin 1%, penicillin 100 U/mL, and streptomycin 50 µg/mL), before long-term storage at −80 °C. The lists of samples are found in Appendix A.

### 2.3. RNA Extraction

For each brain sample, a piece of approximate 0.3 cm^3^ was excised and placed into a 1.5 mL nuclease free plastic tube (Thermo Fisher Scientific, Shanghai, China) containing four or five glass beads with a diameter of 3.5 mm (Sigma, St. Louis, MO, USA). A volume of 1 mL TRIzol reagent (Invitrogen, Shanghai, China) was added and the sample was disrupted at 30 rpm/min for 2 min with TissueLyser II (Qiagen, Shanghai, China). After addition of 0.2 mL of chloroform, vortexing, and incubation at room temperature for 2–15 min, centrifugation at 12,000× *g* for 15 min at 4 °C was performed. The upper phase was collected, added with 0.5 mL of isopropanol, and incubated for 5–10 min at room temperature after mixing. RNA was then precipitated after centrifugation at 12,000× *g* for 8 min at 4 °C, and washed with 1 mL of 75% ethanol solution. After another centrifugation of 5 min at 7500× *g* at 4 °C, the pellet was air dried for 30 min to 1 h at room temperature, resuspended in 50 μL of nuclease free water (Ambion, Shanghai, China), and incubated for 10 min at 60 °C. For saliva swab samples, pools of five samples (40 μL each) were similarly extracted (without the disrupting step) using 0.8 mL of TRIzol LS (Invitrogen), adding 2 μL of glycogen (5 mg/mL) (Ambion) during the isopropanol precipitation step. All RNA samples were stored in −80 °C for subsequent analysis.

### 2.4. Identification of Bat Species

Bat species was determined on morphological features, and doubtful or unidentified species were identified after partial sequencing of the cytochrome b (*CYTB*) gene. Briefly, amplification of an approximate 1180 bp fragment of the *CYTB* gene was performed by polymerase chain reaction (PCR) using forward and reverse primers L14724ag (5′-ATGATATGAAAAACCATCGTTG-3′) and H15915ag (5′-TTTCCNTTTCTGGTTTACAAGAC-3′), respectively, as previously described [49]. PCR products were Sanger sequenced and contigs from assembled sequences were analyzed by BLASTn using the NCBI database. Identification of host species was done on the most closely related sequences with the highest query coverage and a minimum identity of 95%.

### 2.5. Combo RT-qPCR for Lyssavirus Screening

Extracted RNA from bat brain and saliva samples was screened for lyssavirus with a previously described dual combined real-time reverse transcription polymerase chain reaction (combo RT-qPCR) method [50]. This technique includes two complementary technologies: a probe-based (TaqMan) RT-qPCR for rabies virus (RABV) detection (pan-RABV RT-qPCR) and an intercalating dye (SYBR Green) based RT-qPCR to detect other lyssavirus species (pan-lyssa RT-qPCR). Both of the two techniques amplify an approximate 120 bp conserved region of the polymerase gene. TaqMan RT-qPCR and SYBR Green RT-qPCR were performed with Superscript III Platinum One-Step qRT-PCR Kit (Invitrogen) and Superscript III Platinum SYBR Green One-Step qRT-PCR Kit (Invitrogen), respectively, according to Dacheux et al. (2016) [50]. For each assay, the RABV CVS strain (RNA extracted from virus suspension) and the European bat lyssavirus 1 (EBLV-1) 8918FRA strain (in vitro synthesized RNA from a plasmid encoding the viral target) were used as positive controls for the TaqMan RT-qPCR and SYBR Green RT-qPCR, respectively. Negative control (nuclease free water) was also included.

In addition, the sensitivity (limit of detection) of the SYBR Green RT-qPCR was evaluated for one representative (isolate D170001) of Yinshui bat virus (YSBV). Briefly, the viral target that includes the region used for detection (partial L gene sequence, positions 1641–2050 bp) was amplified using primers D170001_pan-lyssa_target_F1 (5′-TCTAATCAAGACCCATTTTG-3′) and D170001_pan-lyssa_target_R1 (5′-GCAAATTGACAATGCT CCAT-3′). Amplification was performed using TaKaRa Ex Taq kit (Takara Bio, Saint-Germain-en-Laye, France) with 5 μL 10X Ex Taq Buffer, 4 μL dNTP Mixture (2.5 mM each), 1 μL of each primer (10 μM), 0.25 μL TaKaRa Ex Taq (5 units/μL), 36.75 μL nuclease-free water, and 2 μL of sample. The amplification was performed as follows: 98 °C for 2 min followed by 35 cycles consisting of 98 °C for 10 s, 47 °C for 30 s, and 72 °C for 30 s, ending with 72 °C for 7 min. The 410 bp length amplicon was cloned into pCRII-TOPO Vector (Invitrogen, Illkirch, France). Recombinant plasmids were amplified in *E. coli* TOP10 and purified with NucleoSpin Plasmid, Mini kit for plasmid (Macherey-Nagel, Hoerdt, France). After confirmation of the presence of the insert by Sanger sequencing, positive clones were quantified by Nanodrop (Thermo Fischer Scientific) to determine the number of target copy per mL. Finally, serial dilutions of the plasmids in nuclease free water were used as standard curves.

### 2.6. Confirmation of Doubtful Samples by Sanger Sequencing

For each doubtful sample, amplicons were cloned using pGEM^®^-T Easy Vector Systems (Promega, Madison, WI, USA) according to the manufacturer’s instructions. A sample was considered as doubtful after comparison of the value and the shape of the melting temperature (Tm) curve with those obtained with the controls, after SYBR Green RT-qPCR analysis. Between 5 and 10 clones for each sample were submitted for Sanger sequencing, and contigs were subsequently analyzed by BLASTn and BLASTx using the NCBI database.

### 2.7. Determination of Rhabdovirus Genome Sequences

The genome sequences of the six new rhabdoviruses (isolates 958, 1127, 1017, D170001, D170022, and D170190) were determined by next generation sequencing (NGS), as previously described [43,51]. Host rRNA depletion was carried out with the kit Terminator 5′-Phosphate-Dependent Exonuclease (Epicentre Biotechnologies, Madison, Wi, USA). Between 2 and 4 μg of extracted RNA (from bat brain) was incubated for 1 h at 30 °C with 2 μL of buffer A and 1 μL of Terminator enzyme, in addition to 0.5 μL of RNAsin Ribonuclease inhibitor (Promega, Charbonnières-les-Bains, France), and adjusted to 20 μL with nuclease free water. Depleted RNA was purified using Agencourt RNAclean XP beads (Beckman Coulter, Villepinte, France) at a ratio of 1.8, as recommended by the manufacturer, and eluted in 20 μL nuclease free water. Eight microliters of purified RNA were then reverse transcribed in complementary DNA (cDNA) using random hexamers (Invitrogen, Illkirch, France) and Superscript III reverse transcriptase (Invitrogen), according to manufacturer’s instructions. Afterward, double-stranded DNA (dsDNA) was synthesized at 16 °C for 2 h in 80 μL final reaction mixture, including 20 μL of cDNA, 8 μL of 10× Second-Strand Reaction Buffer (New England Biolabs, Evry, France), 3 μL of dNTP mix (10 mM) (Invitrogen), 1 μL (10 U) of *E. coli* DNA ligase (New England Biolabs), 4 μL (40 U) of *E. coli* DNA Polymerase I (New England Biolabs), 1 μL (5 U) of *E. coli* RNase H (New England Biolabs), and 43 μL of nuclease free water. Finally, dsDNA was purified using AMPure XP (Beckman Coulter) at a ratio of 1.8, as recommended by the manufacturer, and eluted in 20 μL nuclease free. The dsDNA libraries were constructed using the Nextera XT kit (Illumina, Evry, France) and sequenced using a 2 × 150 nucleotide paired-end strategy on the NextSeq500 platform.

NGS data were analyzed using de novo assembly and mapping, as previously described [43]. Briefly, pre-processing of reads, de novo assembly, and mapping (both using CLC Assembly Cell, Qiagen) were performed with a dedicated workflow implemented on the Pasteur Galaxy platform [52] and slightly modified from Troupin et al. (2016) [53] (workflow available on request). In parallel, another dedicated de novo assembly workflow (named Viroscreen), was performed according to Dacheux et al. (2019) [43], after adaptation from Dacheux et al. (2014) [54]. The viral sequence contigs were assembled and manually edited to produce the final sequences of the viral genomes using Sequencher 5.2.4 (Gene Codes Corporation, Ann Arbor, MI, USA). The quality and accuracy of the final genome sequences were checked after a final mapping step of the original cleaned reads and visualized using Tablet [55].

In the case of low covering regions or the presence of gaps, different primer datasets were designed, based either on the sequences obtained by NGS (specific primers) or on the most closely related sequences available in GenBank (Appendix A). These primers were used by conventional PCR/nested PCR, and amplicons were submitted for Sanger sequencing. Contig sequences were then used to fulfil the gaps into the genome sequences.

The 5′ ends of the viral genomes were obtained or confirmed by rapid amplification of cDNA ends (RACE) with the kit SMARTer RACE 5′/3′ (Takara, Beijing, China). The 3′ race ends were connected using a phosphorylated end by T4 RNA ligase (Takara) and amplified by hemi-nested PCR, as previously described [56]. Sequences were assembled and manually edited to produce final sequences of the viral genome with Sequencher 5.2.4.

Complete genome sequences were deposited in GenBank under the accession numbers MN607592–MN607597.

### 2.8. Analysis of Sequences

Putative open reading frames (ORFs) were identified by Sequencher 5.2.4, and the presence of accessory genes was evaluated in silico after comparison with similar genome regions available in GenBank of other rhabdoviruses belonging to the same genus.

Phylogenetic analysis of the newly rhabdoviruses was conducted using reference genome sequences of other rhabdoviruses available and downloaded from GenBank (Appendix A). For each virus, ORFs were concatenated and amino acid sequence alignments were performed using ClustalW 2.0 [57]. Phylogenetic tree was constructed with MEGA7.0 [58] and PhyML3.0 [59], using the maximum likelihood method with 1000 bootstraps. Amino acid sequence identity of the viral polymerase was determined by MEGA 7.0.

### 2.9. Specific RT-qPCR for Rhabdovirus RNA Quantification in Infected Bats

Rhabdovirus RNA quantification in the different organs of the positive bats was performed by RT-qPCR using HiScript II One Step qRT-PCR SYBR Green Kit (Vazyme, Nanjing, China) following the manufacturer’s recommendations. The amplification was performed as follows: 50 °C for 15 min, 95 °C for 5 min, followed by 40 cycles consisting of 95 °C for 10 s and 60 °C for 30 s. A melting curve analysis was done at 60 °C. Specific primers were designed using PrimerExpress3.0.1 software (Applied Biosystems, Shanghai, China) and amplified an approximate 60 bp region of the polymerase gene. Primers 958-L-660F (5′-CAAGAATGATGATAGAGCGACACA-3′) and 958-L-728R (5′-TTTCAGGTGGGAAATGTTTATTGA-3′) were designed for isolate 958 (putative TYBV species); primers DB170001-L-1180R (5′-GTTGCAGCAGGCGTGGTAT-3′) and DB170001-L-1240F (5′-TTTTAAACACTGCGCGAACAGT-3′) for isolates 1017, D170001, and D170022 (putative YSBV species); and primers 1127-L-1421R (5′-CTACCATTCCTCACCCCCTAGA-3′) and 1127-L-1484F (5′-CCGAGACCTATCCCTCCACTT-3′) for isolate 1127 (putative YSBV species). Absolute RNA quantification was performed using specific standard curves. Standards were produced after amplification by conventional PCR of a fragment of nearly 500 bp in the polymerase gene (containing the region used for quantification) from samples 958, 1127, and D170001, which was subsequently cloned into pGEM-T Easy Vector (Promega). Primers’ sequences used for the production of standards were 958-L-378F (5′-CAAGAATGATGATAGAGCGACACA-3′) and 958-L-1169R (5′-ATCTGGGAAACAAAGGTC-3′) for isolate 958, DB170001-981F (5′-AATGAGCCCATCCAACTT-3′) and DB170001-1467R (5′-TAATCCAATGCCTCCACT-3′) for isolate DB170001, and 1127-L-1043F (5′-TTCAGAGCCGACATACAA-3′) and 1127-L-1537R (5′-TAAGACCACCAACGCAAT-3′) for isolate 1127. Amplification was performed using the Platinum Taq DNA Polymerase kit (Invitrogen, Shanghai, China), with 2.5 μL 10× PCR Buffer, 1 μL MgCl_2_ (50 mM), 0.5 μL dNTP Mixture (10 mM), 1 μL of each primer (10 μM), 0.1 μL Platinum Taq DNA Polymerase (10 units/μL), 17.9 μL nuclease-free water, and 2 μL of sample. For all sets of primers, amplification was performed as follows: 94 °C for 2 min followed by 40 cycles consisting of 94 °C for 20 s, 48 °C for 30 s, and 72 °C for 1 min, ending with 72 °C for 5 min. After cloning, recombinant plasmids were amplified in *E. coli* DH10B and purified with EndoFree Mini Plasmid Kit II (Tiangen Biotech, Beijing, China). In vitro RNA synthesis was carried out by MAXscript Kit (Ambion) for each recombinant plasmid, and then freshly transcribed RNA was purified by TRIzol (Invitrogen) and quantified by Nanodrop (Thermo Fischer Scientific, Shanghai, China) to determine the number of target copy per mL. Finally, serial dilutions of RNA in nuclease free water were used as standard curves.

### 2.10. Virus Isolation

Different cell lines were used for virus isolation: C6-36 (ATCC CRL-1660), Vero (ATCC CCL-81), Vero E6 (ATCC CRL-1586), and *Rhinolophus sinicus* cell lines (brain, lung, liver, kidney, and heart) [60]. Cells’ monolayers were maintained for 80% confluency in DMEM medium (Gibco, Thermo Fisher, USA) supplemented with 10% fetal calf serum (FCS) (Gibco, Thermo Fisher) in 24-well plates at 37 °C with 5% CO_2_, and then washed with serum free DMEM (sfDMEM) after removing the medium. Approximately 0.3 cm^3^ of crushed brain samples (in 200 μL sfDMEM) was centrifuged at 10,000× *g* for 10 min at 4 °C. The supernatant was diluted at 1:10 in sfDMEM before being added to the cells. After 1 h incubation at 37 °C with 5% CO_2_, the inoculum was removed and replaced with fresh DMEM medium containing 2% FCS. The cells were incubated at 37 °C with 5% CO_2_ and checked daily for cytopathic effect. After three blind passages, the cells were harvested and total RNA extraction was performed using TRIzol reagent. Detection of rhabdovirus RNA was realized with the specific RT-qPCR developed for viral RNA quantification.

In parallel, virus isolation was attempted in vivo, using newborn BALB/c mice (3 days old) (Charles river, Beijing, China). Samples were diluted at 1:10 in sfDMEM, and then centrifuged at 10,000× *g* for 10 min at 4 °C. Approximately 5–10 μL of each sample was inoculated intracerebrally to 4–5 newborn mice. The injected animals were monitored daily and euthanized at day 5 and day 7 after inoculation. Brains were collected and total RNA was extracted by TRIzol reagent. Detection of rhabdovirus RNA was realized with the specific RT-qPCR developed as previously described.

## 3. Results

### 3.1. Bat Brain and Saliva Sample Collection

A total of 4576 samples, including bat brain (*n* = 1044) and saliva swab (*n* = 3532) samples, were collected between 2006 and 2018 in 21 provinces of China from 63 bat species representing 23 genera belonging to 8 families (Figure 1, Table 1 and Appendix A).

The four most represented Chinese provinces were Guangdong (*n* = 1285; 28.1%), Yunnan (*n* = 1210; 26.4%), Guangxi (*n* = 893; 19.5%), and Hubei (*n* = 582; 12.7%), all together encompassing 3970 bat samples (72.5%) (Figure 1, Appendix A). Among the eight different bat families sampled, all excepted one (*Pteropidae* family, 5 species) were associated with insectivorous bats (58 species), with the highest number of samples associated with the *Vespertilionidae* family (*n* = 1388; 30.3%), followed by the *Rhinolophidae* (*n* = 1243; 27.2%) and *Hipposideridae* (*n* = 946; 20.7%) families (Table 1 and Appendix A). At the bat species level, the most abundant samples were collected from *Rhinolophus sinicus* (*Rhinolophidae* family), with 547 samples (12%), and from *Hipposideros armiger* (*Hipposideridae*), with 399 samples (8.7%) (Appendix A). Only a limited part of the bats was sampled for both brain and saliva samples. The total number of paired samples was 144 (3.1%) (8 species including 2 frugivorous bats species), which were collected in Hubei (30 samples, 2 species) and Yunnan (114 samples, 6 species) provinces from 2016 to 2018 (Appendix A).

### 3.2. Bat Sample Screening and Description of Positive Specimens

All the saliva swab samples (*n* = 3532) tested with the combo RT-qPCR were found to be negative (Appendix A). Similarly, none of the 1044 bat brain samples were detected as positive with the pan-RABV RT-qPCR, but six samples (0.6%, with samples 958, 1017, 1127, D170001, D170022, and D170190) were classified as doubtful with the pan-lyssa RT-qPCR (Appendix A). Each amplicon (nearly 120 bp) was cloned and Sanger sequenced, and BLASTn/BLASTx analysis indicated the presence of rhabdovirus polymerase sequences. The ability of the combo RT-qPCR, and more specifically of the pan-lyssa RT-qPCR (SYBR Green), to detect rhabdoviruses other than lyssaviruses is explained by the targeted viral region, corresponding to a relatively conserved region within the polymerase gene of this virus family (Appendix A) [50]. However, despite this relatively high level of conservation and depending on the sequence of the rhabdovirus considered, the presence of mutations in the primers regions probably impacts the sensitivity of detection, as exemplified with the high limit of detection (1 × 10^7^ copies per assay) for isolate D170001 (Appendix A).

All six were detected in bats belonging to the *Rhinolophus* genus and originating from two provinces: Hainan and Hubei (Figure 1, Table 1). Sample 1127 was obtained from *Rhinolophus affinis* in Qiongzhong city of Hainan province in August 2007. The five other samples were collected from bats belonging to the species *Rhinolophus sinicus* collected in Xianning city in Hubei province. Samples 958, 1017, and D170190 were collected in the Taiyi cave in May 2007 (samples 958 and 1017) and March 2017 (sample D170190), whereas samples D170001 and D170022 were collected in the Yinshui cave, 34 km from the Taiyi cave, in March 2017.

The detection rate of rhabdovirus in these three locations was evaluated for both *Rhinolophus* species (*sinicus* and *affinis*) during the period of the study (2006–2018) (Table 2). This overall detection rate was 11.1% (1/9) in Hainan province for *Rhinolophus affinis*, and 2.8% (3/107) or 4% (2/50) for *Rhinolophus sinicus* in Hubei province in Taiyi or Yinshui caves, respectively. Over the period, the overall detection rate in Hubei province was 3.2% (5/157). Interestingly, the positivity rate for Taiyi cave, which has been sampled 10 times over the study period, was 8.3% (2/24) in 2007 and 7.1% in 2017 (1/14) (Table 2). Remarkably, positive samples from Yinshui cave were obtained during the unique collection done in 2017 (overall detection rate of 4.7% (3/64) in Hubei province). However, the number of positive samples or total samples tested remained low.

### 3.3. Genome Characterization of Chinese Bat Rhabdoviruses

Determination of the nucleotide genome sequences of the six bat rhabdoviruses detected by combo RT-qPCR was initially done by NGS, generating 10 million reads in average per sample. Remaining gaps and low coverage regions were resolved by specific PCR or nested-PCR and Sanger sequencing of the corresponding amplicons (directly or after cloning). Determination of genome extremities was performed using RACE approaches. Based on NGS and/or Sanger sequencing, nearly complete genome sequences were obtained for all six samples, ranging from 10,868 to 10,952 nt in length (Figure 2). The 5′ trailer sequence was obtained for all except for isolates 958, whereas the 3′ leader sequence was not successfully recovered, even after several attempts. BLASTn and BLASTx analyses performed on the genome sequences demonstrated that all these viruses were most closely related to the *Vesiculovirus* genus, except isolate 958, which was found to be more related to the *Ledantevirus* genus. This taxonomic classification was further confirmed by phylogenetic analysis (see Section 3.4).

All six genomes exhibited a typical rhabdovirus organization, consisting of the five canonical genes encoding, in the following order, the nucleoprotein (N), the phosphoprotein (P), the matrix protein (M), the glycoprotein (G), and the polymerase (L) (Figure 2).

The genomic characteristics of the five isolates (1127, 1017, D170001, D170022, and D170190) related to vesiculoviruses were highly similar. Indeed, each of the N, M, and L proteins were identical in size, with 423 aa (1269 nt), 209 aa (627 nt) and 2105 aa (6315 nt), respectively. Only isolate 1127 exhibited differences in the length of the P (241 aa, 723 nt) and G (521 aa, 1563 nt) proteins compared with the four other isolates, with 255 aa (765 nt) and 513 aa (1539 nt). The transcription initiation (TI) signal was highly conserved, with the TARCAGR sequence (R = A/G), whereas the consensus sequence for the transcription termination (TTP) was YMTGA_7_ (Y = C/T and M = A/C) (Appendix A). In addition, various potential ORFs representing putative accessory genes (U) (cut off > 140 nt) were identified (from 4 to 5, depending on the isolates) (Figure 2). The ORFs coding for the putative proteins U2 (98 aa, 294 nt), U3 (52 aa, 156 nt), and U4 (65 aa, 196 nt) were found in the P, M, and G genes of the genome for the isolates 1017, respectively, and similar putative proteins were found in the same regions of the genomes for the isolates D170001, D170022, and D170190. Relatively similar ORFs in the P and G genes were observed for isolate 1127, with U2 (50 aa, 150 nt) and U5 (85 aa, 258 nt), respectively. The sequence of the 5′ trailer region was 56 nt in length and identical for isolates 1127, 1017, D170001, D170022, and D170190.

The isolate 958, related to ledanteviruses, presented the shortest genome (10,933 nt) among this genus (excluding the leader sequence, which was not available). The N, P, M, G, and L proteins were 428 aa (1284 nt), 283 aa (849 nt), 194 aa (582 nt), 536 aa (1608 nt), and 2121 aa (6363 nt) in length, respectively. Interestingly, the M protein of this isolate was the shortest compared with other members of the *Ledantevirus* genus (208–217 aa, 624–651 nt), and the G protein was the second shortest after the G protein of Mount Elmont bat virus (MEBV) (535 aa, 1605 nt). The consensus sequences of the TI and TTP signals were AACGWGW (W = C/G) and CRTGA_7_ (R = A/G), respectively (Appendix A). Only one additional ORF (U1) was found in the N gene, which putatively encoded a 48 aa protein (144 nt). The length of the 5′ trailer region was uncompleted (32 nt missing).

### 3.4. Sequences Analysis of Chinese Bat Rhabdoviruses

Phylogenetic analysis of the six Chinese bat rhabdoviruses was conducted on both the complete amino-acid sequences of the L protein and the nucleotide genome sequences, with representative members of the *Rhabdoviridae* family.

Maximum likelihood phylogenetic analysis performed on the conserved and complete L proteins of these Chinese bat rhabdoviruses, with sequences of 155 other rhabdovirus members available in GenBank (Appendix A), demonstrated that isolates 1127, 1017, D170001, D170022, and D170190 were putative new members of the *Vesiculovirus* genus, whereas the isolate 958 clustered within the *Ledantevirus* genus (Figure 3). These results, strongly supported by bootstrap values, confirm the preliminary classification based on BLAST analysis. Interestingly, all the new Chinese bat vesiculoviruses clustered with other vesiculoviruses previously found in bats, with American bat vesiculovirus (ABV) from North America [30] and the recently described Jinghong bat virus (JHBV) originating from the same bat species (*Rhinolophus affinis*) in Yunnan province, China [29]. Within this cluster, the four isolates 1017, D170001, D170022, and D170190 formed a monophyletic group and were considered as a single putative new rhabdovirus species, tentatively named Yinshui bat virus (YSBV) (Figure 3). The isolate 1127, tentatively identified as Qiongzhong bat virus (QZBV) species, did not fall into this cluster, but was clustered with the Chinese vesiculovirus JHBV. Lastly, the isolate 958, which was tentatively classified as Taiyi bat virus (TYBV) species, was also associated with other bat-related members within the *Ledantevirus* genus, and more specifically with Mount Elgon bat virus (MEBV) (Figure 3) [61].

These results were confirmed by phylogenetic analysis conducted on the nucleotide genome sequences (Figure 4). Indeed, similar topologies of the tree were found for YSBV (isolates 1017, D170001, D170022, and D170190) and QZBV (isolate 1127), which clustered within the *Vesiculovirus* genus (Figure 4a), and for TYBV (isolate 958), which was strongly related to *Ledantevirus* genus and more precisely to MEBV (Figure 4b).

The phylogenetic relation between the six novel bat rhabdoviruses and the other members within the same genus (i.e., *Vesiculovirus* and *Ledantevirus*) was supported by high amino acid identities of each canonical proteins (Appendix A). Within the genus *Vesiculovirus*, the N and L proteins were the most conserved for QZBV species (isolate 1127), with amino acid identities ranging from 43% to 100% and 52% to 91%, respectively, as well as for YSBV species (isolates 1017, D170001, D170022, and D170190), ranging from 44% to 90% and 52% to 76%, respectively (Appendix A). Identically, both of these proteins were the most conserved for the genus *Ledantevirus*, with amino acid identities ranging from 38% to 87% for TYBV species (isolate 958) and from 48% to 76% for N and L proteins respectively (Appendix A). In contrast, the P protein was the most variable for YSBV (amino acid identities from 9% to 52%), QZBV (11% to 87%), and TYBV (14% to 74%) (Appendix A).

Based on sequence comparison and phylogenetic analysis, the four isolates 1017, D170001, D170022, and D170190 exhibited a high homology in the genome sequences, suggesting that they belong to the same potential new species YSBV (Figure 3 and Figure 4a). Indeed, only 109 nucleotide mutations were found after genome sequence comparison, with 21 mutations associated with amino acid residue modification (Figure 5). For isolates D170001 and D170022, isolates that have been collected from *Rhinolophus sinicus* in Yinshui cave in 2017, a single amino acid (G gene) residue mutation was observed between them, in addition to three synonymous mutations in other parts of the genome. In parallel, only 7 amino acids mutations (four mutations in the P gene, two mutations in the G gene, and one mutation in the L gene) and 26 synonymous mutations were identified between isolates 1017 and D170190, both collected in Taiyi cave from the same bat species in 2007 and 2017, respectively (Figure 5). A total of 21 amino acids mutations were found between D170001/D170022 and 1017/D170190 in P, M, G, and L genes (Figure 5).

### 3.5. Organ Tropism of the Novel Chinese Bat Rhabdoviruses

Specific RT-qPCRs were designed to quantify the viral RNA concentration of the six novel rhabdoviruses in different organ samples from the original infected bats. Depending on the specimens, several tissues were tested, including brain, heart, liver, spleen, lung, kidney, and intestine (Table 3). For all rhabdoviruses, the viral concentration was high, between 1.12 × 10^8^ copies/g and 1.62 × 10^12^ copies/g, regardless of the sample tested. The only exception was the concentration of viral RNA found in intestine for isolate 958 (4.68 × 10^5^ copies/g). These results demonstrated that multiple organs can be infected, with a high viral load. This is especially true for YSBV species with isolates D170001 (seven different organs tested) and D170190 (seven organs tested) and, to a lesser extent, for isolates 958 and 1017 (two organs tested with brain and intestine) (Table 3).

### 3.6. Bat Sample Screening by Specific RT-qPCRs

The three RT-qPCRs specifically designed to detect and quantify TYBV (isolate 958), QZBV (isolate 1127) and YSBV (isolates 1017, D170001, D170022, and D170190) viruses were used on different bat samples available from *Rhinolophus sinicus* and *Rhinolophus affinis*, collected in the same location as the positive bat specimens (Yinshui and Taiyi caves in Hubei province, Qiongzhong in Hainan province). A total of 394 samples were tested, including samples already analyzed by the combo RT-qPCR and additional samples (Appendix A). All of the 160 brain samples, 29 saliva and 205 feces were found to be negative.

### 3.7. Virus Isolation

Attempts to isolate the six different new bat rhabdoviruses were conducted with different organs from the original infected specimens, on cell culture with different cell lines, including C6-36 (ATCC CRL-1660), Vero (ATCC CCL-81), Vero E6 (ATCC CRL-1586), and *Rhinolophus sinicus* cell lines (brain, lung, liver, kidney, and heart), without any visible cytopathogenic effects nor positive RNA detection with specific RT-qPCR. Similarly, virus isolation after intracerebrally inoculation of crushed tissues in newborn mice did not lead to mortality or any clinical symptoms, or to the detection of viral RNA in inoculated brain after euthanasia.

## 4. Discussion

It is now clearly demonstrated that bats are major reservoirs of viruses, and could also act as vectors of emerging or re-emerging viral zoonosis [4,5,62,63]. Among them, rabies virus was one of the first identified in these animals [64]. Indeed, rabies virus and other members of the *Lyssavirus* genus have been described in several part of the world in bats, including the Americas (with the unique presence of *Rabies lyssavirus* species), Africa (with at least three different species of bat related-lyssaviruses—*Duvenhage lyssavirus*, *Lagos bat lyssavirus*, and *Shimoni bat lyssavirus*—and two potential ones—*Ikoma lyssavirus* and *Mokola lyssavirus*), and Europe (five different species described—*Bokeloh bat lyssavirus*, *European bat 1 lyssavirus*, *European bat 2 lyssavirus*, *Lleida bat lyssavirus*, and *West Caucasian bat lyssavirus*—and one tentative species with Kotalahti bat lyssavirus) [22]. Only one bat lyssavirus has been described so far in Australia and in the Indian subcontinent—*Australia bat lyssavirus* and *Gannoruwa bat lyssavirus*, respectively. Excepted for Middle Asian regions, where three different bat lyssavirus species were discovered (*Aravan lyssavirus*, *Irkut lyssavirus*, and *Khujand lyssavirus*), the other parts of Asia, and more especially China, remain largely unexplored. To date, only two bat lyssavirus species have been identified—*Taiwan bat lyssavirus* in two Japanese pipistrelles (*Pipistrellus abramus*) in Taiwan in 2017 [46] and with *Irkut lyssavirus* in a single greater tube-nosed bat (*Murina leucogaster*) bat in Tonghua county, Jilin province in 2012 [47]. In addition, a single rabies seroprevalence study has been conducted by indirect fluorescent antibody test in bats sampled in South of China during the period 2005–2006, and identified a low rabies seroconversion: 2.2% (15/685) [48].

In this context, the aim of our study was to fill this gap by improving the knowledge on the prevalence of lyssavirus infection in Chinese bat. For this purpose, we realized one of the largest molecular screenings of retrospective bat samples (*n* = 4576), with 1044 brains and 3532 saliva collected during the period 2006–2018 from 63 different bat species originating from 21 provinces of China. As at least 135 different bat species were reported in this country [65], our representative dataset covered nearly 50% of the bat species diversity found in 62% of the country (21 out of the 34 existing Chinese provinces). We applied a validated technique for pan-lyssavirus detection (combo RT-qPCR), based on two complementary technologies: a probe-based (TaqMan) RT-qPCR for rabies virus (RABV) detection (pan-RABV RT-qPCR) and an intercalating dye (SYBR Green) based RT-qPCR to detect other lyssavirus species (pan-lyssa RT-qPCR) [50]. None of the samples tested were found to be positive for rabies virus, nor other lyssaviruses. This result was not surprising because most of the samples were collected on apparently healthy and flying bats with nets when they left their roost caves at dusk, or directly in the colonies with landing nets [66]. Indeed, multiple lines of evidence demonstrated that bats are sensitive to lyssavirus infection, which can induce signs of rabies (including inability to fly, abnormal behavior, aggressiveness) and ultimately lead to death [22,25,67], even if we cannot exclude asymptomatic infection. All new bat lyssavirus species have been detected in sick or dead animals [22,46,68,69], and the proportion of positive samples has been demonstrated to be higher when targeting this category of animals within bat colonies [70]. On the other hand, in such a healthy bat population, previous studies demonstrated that the positive rate for lyssavirus detection was expected to be very low, around 0.1% in the brain or saliva sample [71,72,73]. However, although the number of brains tested (nearly 1000 samples) may explain the lack of positivity, we would have expected to identify around three positive saliva samples from our dataset (nearly 3500 samples), based on this positive detection rate of 0.1%. One reason could rely on the dilution factor associated with the storage medium used during saliva sample collection and the extraction step. Indeed, saliva swabs were immersed in 1 mL VTM after collection, and only 40 μL was used for RNA extraction. This dilution factor (1:25) could decrease the overall sensitivity of detection. Usually, similar studies are conducted on saliva samples collected in a smaller volume (nearly 200 μL of VTM, TRIzol, or other) and RNA was extracted from the entire volume [71,72]. Another limitation of this study is due to the undersigned sampling. Although our sampling periods cross more than 12 years, the tissue samples were randomly collected in some bat populations with high density and from bats of accidental death during sampling, explaining the sampling disparities at the geographic temporal and bat species level (Figure 1, Appendix A).

Despite the absence of lyssavirus detection, we were able to detect the presence of other rhabdoviruses with the pan-lyssa RT-qPCR in six bat brain samples (0.6%), with samples 958, 1017, D170001, D170022, and D170190. Among these six positive brain samples, all collected from bats belonging to *Rhinolophus* genus, we were able to characterize one putative new species (tentatively named Taiyi bat virus or TYBV) of the *Ledantevirus* genus with isolate 958, and clustered within two potential other species of the *Vesiculovirus* genus, with Yinshui bat virus (YSBV) and Qiongzhong bat virus (QZBV) species for isolates 1017, D170001, D170022, and D170190, and for isolate 1127, respectively. The extension of the spectrum of detection of this assay, initially dedicated to lyssavirus, rely on the use of a relatively conserved region on the L gene (positions 7272–7390 according to the reference sequence PV- GenBank accession number M13215) as a target (Appendix A) [50]. Previous studies based on molecular techniques targeting similar conserved regions on this gene reported similar broad detection among the *Rhabdovirus* family [44]. Despite this level of conservation, the presence of potential mutations in these regions, depending on the rhabdovirus species considered, could negatively impact the sensitivity of this technique, and lead to not detecting animals with low viral loads (Appendix A). It should, however, be noted that the use of RT-qPCRs specific for the new bat rhabdoviruses confirmed the results obtained on the same samples with the combo RT-qPCR dedicated for lyssavirus detection. Additional screening of samples for *Rhinolophus sinicus* and *Rhinolophus affinis* from the same location/cave (i.e., Taiyi and Yinshui caves, Qiongzhong location) with specific RT-qPCR remained negative (Appendix A).

According to the species demarcation criteria of vesiculoviruses proposed by ICTV, viruses assigned to different species within the genus *Vesiculovirus* have minimum amino acid sequence divergence of 20%, 10%, and 15% in L, N, and G proteins, respectively. YSBV (isolates 1017, D170001, D170022, and D170190) meets these requirements, with amino acid sequence divergence of 24%, 10%, and 27% for L, N, and G proteins, respectively, when compared with the most closely related other vesiculoviruses (JHBV) (Appendix A). QZBV (isolate 1127) is assigned to the same tentative species with the vesiculovirus JHBV, with amino acid sequence divergence of 9%, 0%, and 15% for L, N, and G proteins, respectively (Appendix A).

TYBV (isolate 958) was identified in a *Rhinolophus sinicus* bat collected in 2007 in Hubei province. It represents the first member of the *Ledantevirus* genus described in China, and the second in Asia, with Oita virus (OITAV) described in bat (*Rhinolophus cornutus*) in 1972 in Japan [74]. The seven other bat ledanteviruses were mainly reported in Africa, with the exception of Vaprio virus (VAPV) reported in Italy (Europe), in 2016 in a *Pipistrellus kuhlii* bat species [32], and Kern Canyon virus (KCV) detected in 1956 in California (USA) in a *Myotis yumanensis* bat specimen [75] (Appendix A). Based on phylogenetic analysis, TYBV was most closely related to the Mount Elgon bat virus (MEBV) in the subgroup B of ledanteviruses, which was isolated in Kenya in 1964 in a *Rhinolophus hilderbrandtii eloquens* bat species [61] (Figure 4b). This subgroup was initially suggested to be related to viruses found in bats belonging to *Rhinolophoidea*, whereas subgroup A could encompass viruses detected in bats from the family *Vespertilionidae* [75]. Based on ICTV criteria for species demarcation, TYBV can also be considered as a new species, exhibiting amino acid divergence of 24% and 36% for L and G proteins, respectively, compared with the most closely related MEBV ledantevirus. It significantly exceeds the proposed minimum divergence of 7% and 15% in L and G protein, respectively (Appendix A). Additionally, their accessory ORFs differ from all known species in terms of lengths and locations (Figure 2).

Interestingly, the new tentative species TYBV also clustered with two insect-related viruses. The first one was Wuhan Louse Fly virus 5 (WLFV5), identified in *Hippoboscidae* sp. fly in China in 2013 [76], whereas Kanyawara virus (KYAV) was described in nycteribiid bat flies in Uganda on collared fruit bats (*Myonycteris* sp) in 2010 [31], and more recently on Angolan soft-furred fruit bats (*Lissonycteris angolensis ruwenzorii*) in 2017 [77]. TYBV was also closely related to the recently described Bughendera virus (BUGV), which was identified in one nycteribiid bat fly during the same study on Angolan soft-furred fruit bat in 2017 [77] (Figure 4b). As suggested by these previous studies and by our phylogenetic data, we cannot exclude that TYBV could act as an arbovirus, similar to other members of ledanteviruses belonging to subgroup b (Figure 4b).

QZBV (isolate 1127) and YSBV (isolates 1017, D170001, D170022, and D170190), all identified in *Rhinolophus sinicus* bat species, clustered with a bat-related subgroup among the *Vesiculovirus* genus (Figure 4a), which was divided according to the geographical origin of isolates. QZBV and YSBV were associated with the other Chinese vesiculovirus Jinghong bat virus (JHBV), whereas the American bat vesiculovirus (ABV) from North America was genetically distinct. Within the Chinese vesiculovirus cluster, QZBV was the closest to JHBV, isolated from a related bat species (*Rhinolophus affinis*) in Yunnan province in 2011. Considering that QZBV was detected in Yunnan province, which is more than 1000 km away from Hainnan province, we can assume that this subcluster of vesiculovirus is widely distributed over the country of China, at least in bat *Rhinolophus* genus.

On the other hand, the four isolates of the tentative species YSBV were all discovered in two caves from the same region in Hubei province (Figure 1), which thus represents a unique epidemiological context, both in time and space. Indeed, isolates 1017 and D170190 were found in the Taiyi cave in 2007 and 2017, respectively, suggesting that YSBV in this cave were present (continuously or not) for at least 10 years. In addition, the high level of genetic homology (only seven amino acid mutation) between the two isolates demonstrated a high stability of YSBV in the Taiyi cave over at least a 10-year period (Figure 5). Similarly, isolates D170001 and D170022 identified in Yinshui cave during the same sampling campaign were closely genetically related, with only four mutations (including one non-synonymous position), suggesting that the same virus was circulating in the *Rhinolophus sinicus* bat colony at this time (Figure 5 and Table 1). Most interestingly and despite the close distance between the two caves, 21 amino acid mutations were found between YSBV isolates from Taiyi and Yinshui caves, indicating the presence of two different virus populations. Such markers would represent potential indicators to investigate the circulation of both YSBV isolates and associated bats (here, *Rhinolophus sinicus*) between the two caves, although this hypothesis needs to be confirmed on a larger number of isolates, in addition to the investigation of the circulation of bat individuals between the two caves, notably through ringing and capture/recapture campaigns. Indeed, little is known about the ecological factors of this species of bat, especially regarding their fight distance, interactions between colonies, or potential migration [78].

Based either on complete nucleotide genome sequences (Figure 3) nor on complete amino acid polymerase sequences (Figure 4a), both QZBV and YSBV isolates clustered in a distinct group within the *Vesiculovirus* genus. This phylogroup also encompasses the closely related Chinese bat vesiculovirus JHBV and the more distant American bat vesiculovirus ABV. Even if the number of representative isolates or species remains limited, we can hypothesize that this phylogroup is strongly associated with bats, with a potential geographical clustering (at least between Asian and North American viruses). It will be of interest to investigate the presence of similar bat vesiculoviruses in other parts of the world, especially Europe and Africa. In addition, and similar to TYBV and other ledanteviruses, we cannot exclude at this time the role of arthropods in the life cycle of these bat viruses, in particular, mainly all other vesiculoviruses are considered as arboviruses. Further investigation in order to develop this hypothesis is necessary, in particular through the collection and screening of parasites infecting bats that are present in Taiyi and Yinshui caves.

Interestingly, we were able to demonstrate that different bat rhabdoviruses could circulate simultaneously in the same cave and same bat species. Indeed, we detected YSBV (isolate 1017) and TYBV (isolate 958) viruses in Taiyi cave in different *Rhinolophus sinicus* bats during the same sampling period (May 2007). This observation is not limited to only rhabdoviruses, because other virus families were found at this time in the same bat species, with SARS-like coronaviruses (unpublished data). This illustrates that virus populations can be large and diverse in the same bat colony. In our study, we did not find bat specimens coinfected with different rhabdoviruses (or even SARS-like viruses, data not shown). However, infection with different viruses has been already observed in bats [54,66].

All QZBV, YSBV, and TYBV isolates were initially detected in the brain of infected bats. Additional molecular screening and virus quantification in other available tissues revealed that the infection was multi-organs, such as intestine, lung liver, or heart, associated with a high viral load. The American vesiculovirus ABV was also detected in the lung and liver of infected bats [28], whereas Chinese bat vesiculovirus JHBV was found in the intestine [29]. This systemic infection raises questions about the impact of infected bats, because they did not exhibit specific clinical signs during their capture, with was done using nets to target bats in flight. It would be interesting to know if these viruses lead to asymptomatic infection, or if the animals collected did not yet have these symptoms (although high viral loads were found in various organs). Further investigations will be also necessary to determine how these viruses are transmitted between individuals and, more particularly, if virus shedding can be possible through urine or feces, although a preliminary screening on 205 feces was negative (Appendix A). It would also be important to retest the presence of these viruses in saliva, using less diluted samples. These elements are important to know in order to assess the zoonotic risk of these new viruses. However, it seems unlikely that these bat rhabdoviruses present a high risk to spillover to other animals and humans, especially when considering the low rate of infection and the low opportunity of contact between bat and human. Identification of non or less invasive samples such as saliva, urine, or feces to further investigate these viruses is also a key element in terms of bat conservation.

## Figures and Tables

**Figure 1 viruses-13-00064-f001:**
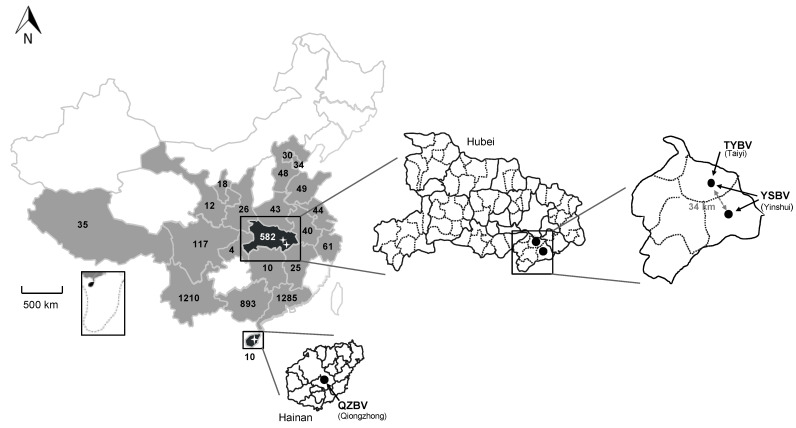
Repartition of the bat samples analyzed in this study and collected in 21 provinces of China (in grey and back) during the period 2006–2018, with the total number of samples indicated per province. The two provinces in black (Hainan and Hubei) indicate the presence of positive samples. Magnifications of these provinces are presented, with the position and the name of the positive sample caves or sites indicated with black dots, as well as the different rhabdovirus identified (TYBV: Taiyi bat virus, YSBV: Yinshui bat virus, and QZBV: Qiongzhong bat virus).

**Figure 2 viruses-13-00064-f002:**
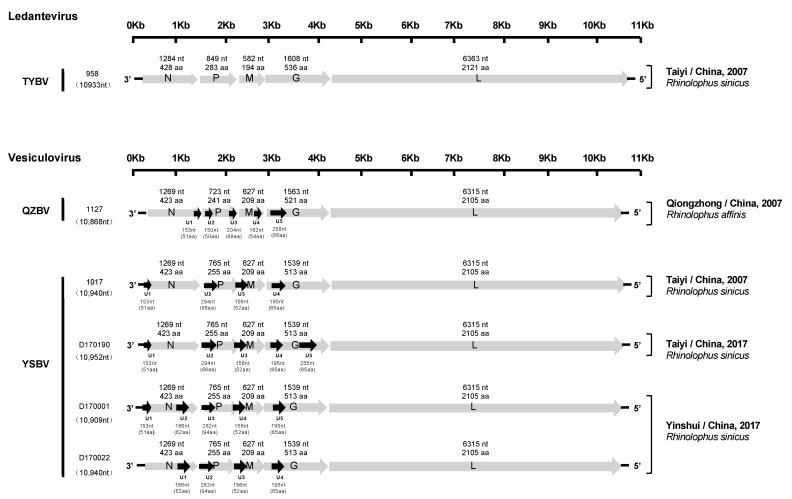
Schematic genomic organization of the six Chinese bat rhabdoviruses. The grey arrows represent the five canonical open reading frames (ORFs) (N, P, M, G, and L), and the black arrows indicate the position of the putative additional ORFs. The nucleotide and amino acid lengths of each ORF are indicated.

**Figure 3 viruses-13-00064-f003:**
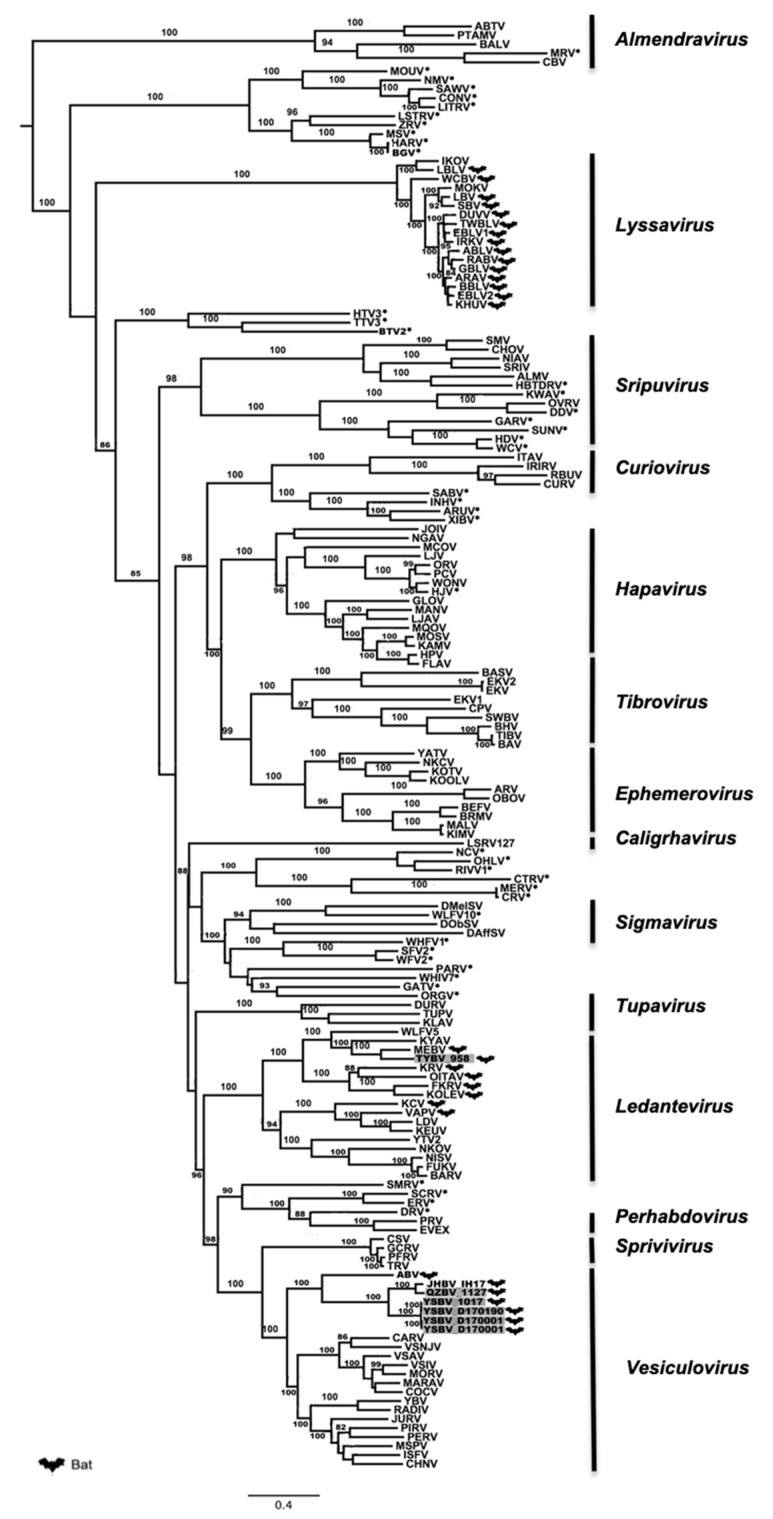
Phylogenetic classification of the sic Chinese bat rhabdoviruses. A maximum likelihood phylogenetic tree was done using MEGA7.0 on the full amino-acid sequence of the L protein including 155 previously reported rhabdoviruses from GenBank, using the LG+G+I+F model and with 1000 bootstrap replicates. Bat-related rhabdoviruses are indicated by a dedicated symbol. Unclassified rhabdoviruses are indicated by asterisks. The Chinese bat rhabdoviruses described in this study are indicated in grey. All bootstrap proportion values (BSP) > 80% are specified. Scale bar indicates nucleotide substitutions per site.

**Figure 4 viruses-13-00064-f004:**
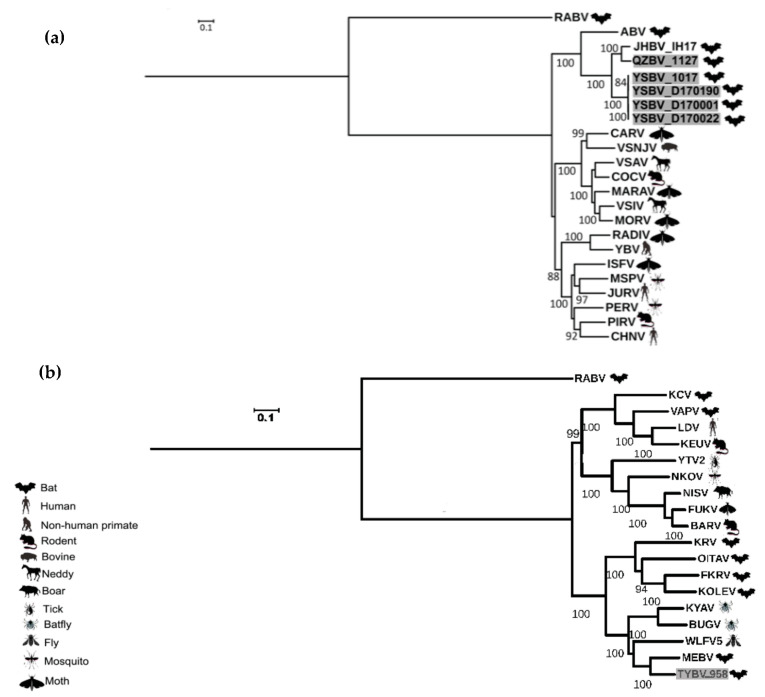
Phylogenetic classification of the six Chinese bat rhabdoviruses. A maximum likelihood phylogenetic tree was done with PhyML3.0 on the nucleotide complete genome sequences, including representative members of the *Vesiculovirus* genus (**a**) and of the *Ledantevirus* genus (**b**), using the GTR+G+I model with a bootstrap of 1000 replicates. The main animal reservoirs for each virus are indicated by specific cartoons and the Chinese bat rhabdoviruses described in this study are indicated in grey. All bootstrap proportion values (BSP) > 80% are specified. Scale bar indicates nucleotide substitutions per site.

**Figure 5 viruses-13-00064-f005:**
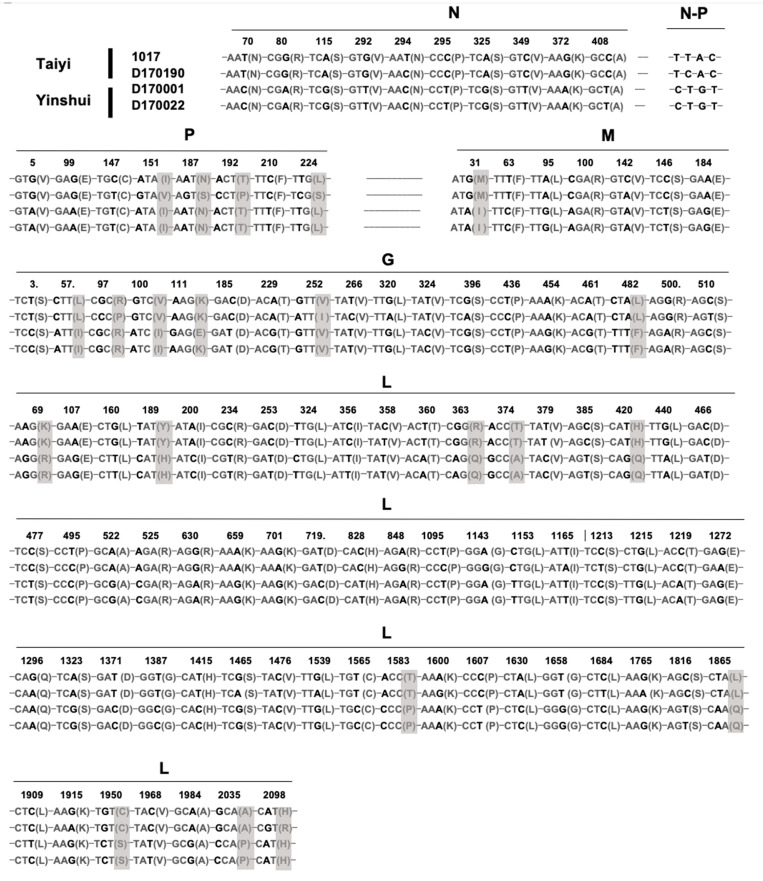
Identification of the nucleotide and amino acid mutations between the four isolates (1017, D170001, D170022, and D170190) of putative YSBV species. The nucleotide mutations are indicated in bold and the amino acid mutations in grey.

**Table 1 viruses-13-00064-t001:** Distribution of the different bat families tested in this study, according to the type of sample.

Bat Family	All Sample		Saliva Sample		Brain Sample	
	No. (%)	Species No.	No. (%)	Species No.	No. (%)	Species No.
*Vespertilionidae*	1388 (30.3)	33	1145 (32.4)	28	243 (23.3)	20
*Rhinolophidae*	1243 (27.2)	14	926 (26.2)	8	317 (30.4)	12
*Hipposideridae*	946 (20.7)	5	689 (19.5)	5	257 (24.6)	5
*Pteropidae* ^1^	345 (7.5)	5	188 (5.3)	4	157 (15)	5
*Miniopteridae*	362 (7.9)	3	293 (8.3)	2	69 (6.6)	3
*Emballonuridae*	210 (4.6)	1	210 (6)	1	0	0
*Megadermatidae*	79 (1.7)	1	78 (2.2)	1	0	0
*Molossidae*	3 (0.1)	1	3 (0.1)	1	1 (0.1)	1
Total	4576 (100)	63	3532 (100)	50	1044 (100)	46

^1^ Bats belonging to this family are frugivorous; all other bat species tested in this study are insectivorous.

**Table 2 viruses-13-00064-t002:** Detection rate of rhabdovirus in brain samples of *Rhinolophus sinicus* and *Rhinolophus affinis* originating from Hubei and Hainan provinces.

	*Rhinolophus affinis* (Hainan, Qiongzhong)	*Rhinolophus sinicus* (Hubei, Taiyi)	*Rhinolophus sinicus* (Hubei, Yinshui)	Total (%)
Date of collection (month/year)	No. positive/No. tested (%)			
08/2006		0/4		0/4 (0)
09/2006		0/11		0/11 (0)
05/2007		2/24 (8.3) *958, 1017* ^1^		2/24 (8.3)
08/2007	1/9 (11.1) *1127*			1/9 (11.1)
04/2009		0/3		0/3 (0)
05/2010		0/1		0/1 (0)
04/2011		0/2		0/2 (0)
03/2017		1/14 (7.1) *D170190*	2/50 (4) *D170001, D170022*	3/64 (4.7)
09/2017		0/10		0/10 (0)
10/2017		0/13		0/13 (0)
05/2018		0/25		0/25 (0)
Total (%)	1/9 (11.1)	3/107 (2.8)	2/50 (4)	6/166 (3.6)

^1^ Identification of positive samples is indicated in italic.

**Table 3 viruses-13-00064-t003:** Virus RNA quantification of the six Chinese bat rhabdoviruses in different organs from the original infected bats.

	Log_10_ Viral RNA Copies per g of Tissue						
Samples	Brain	Heart	Liver	Spleen	Lung	Kidney	Intestine
958	8.06	NA ^1^	NA	NA	NA	NA	5.67
1017	10.47	NA	NA	NA	NA	NA	8.05
D170001	10.76	11.78	11.21	12.21	11.78	11.22	9.06
D170022	10.31	NA	NA	NA	NA	NA	NA
D170190	8.72	9.20	8.87	NA	10.01	9.65	NA
1127	10.22	NA	NA	NA	NA	NA	NA

^1^ NA: not applicable.

## Data Availability

The data presented in this study are available in the present article and in Appendix A. Complete genome sequences were deposited in GenBank under the accession numbers MN607592–MN607597.

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
