# Peer review of "Characterization of Novel Rhabdoviruses in Chinese Bats"

_viruses, 2021, doi:10.3390/v13010064_

Round 1
Reviewer 1 Report
Presented manuscript describes isolation and characterization of novel rhabdoviruses in the population of Chinese bats. It delivers new data concerning rhabdoviruses harboured in Rhinolophus bats. The results suggests simultaneous circulation of different rhabdoviruses in a same cave but not in the same bat individual that should be further studied in the future.
The manuscript is well written with methodological details and the study is properly designed as well as performed. However, some minor changes should be performed before publishing:
Introduction:
- line 56, I suggest to rewrite: rabies virus - RABV
- line 68, should be hepavirus than hepaviruse
- line 75, I suggest replace diffusion by spillover
Materials and Methods:
- Part 2.1. should be carefully check, double usage according to.. should be corrected
- line 140: correct identification European bat lyssavirus 1 not Europe ......
- correct Thermofisher Scientic, USA.
Results:
- Figure 3 - very low quality, difficult to read the text
- line 351: rhabdoviruses than rhabdovirus
- line 494: Check carefully meaning of the sentence - Except in Middle Asia...
Discussion:
Discussion part covers all important aspects of the manuscript. However, I suggest to discuss the differences in the prevalence of rhabdovirus detection in different location (provinces) more deeply (the possible reasons of differences). Temporal distribution of samples would increase the scientiffic value of the manuscript.
Author Response
Reviewer 1
Presented manuscript describes isolation and characterization of novel rhabdoviruses in the population of Chinese bats. It delivers new data concerning rhabdoviruses harboured in Rhinolophus bats. The results suggests simultaneous circulation of different rhabdoviruses in a same cave but not in the same bat individual that should be further studied in the future.
The manuscript is well written with methodological details and the study is properly designed as well as performed. However, some minor changes should be performed before publishing:
Introduction:
- line 56, I suggest to rewrite: rabies virus - RABV
- line 68, should be hepavirus than hepaviruse
- line 75, I suggest replace diffusion by spillover
Correction done.
Materials and Methods:
- Part 2.1. should be carefully check, double usage according to.. should be corrected
- line 140: correct identification European bat lyssavirus 1 not Europe ......
- correct Thermofisher Scientic, USA.
Correction done.
Results:
- Figure 3 - very low quality, difficult to read the text
- line 351: rhabdoviruses than rhabdovirus
- line 494: Check carefully meaning of the sentence - Except in Middle Asia...
Correction done.
Discussion:
Discussion part covers all important aspects of the manuscript. However, I suggest to discuss the differences in the prevalence of rhabdovirus detection in different location (provinces) more deeply (the possible reasons of differences). Temporal distribution of samples would increase the scientiffic value of the manuscript.
We thank the Reviewer 1 for this suggestion. In our study, we mainly focused on the geographical and temporal distribution of the positive samples. Indeed, as added in the Discussion section “Although our sampling periods cross more than 12 years, the tissue samples were randomly collected in some bat populations with high density and from bats of accidental death during sampling, explaining the sampling disparities at the geographic temporal and bat species level (Figure 1, Table S1 and S2).” lines 552-555. Thus, it remains difficult to draw strong conclusions regarding the temporal or geographical distribution of our sampling.
Reviewer 2 Report
Luo et al manuscript, entitled: “Characterization of novel rhabdoviruses in Chinese bats”, describes a molecular retrospective study for detection of lyssavirus in biospecimens collected from bats of Chinese origin. The team investigated 1044 brain and 3562 saliva samples from 63 different bats species. The sample collection span from 2006 to 2018 (with a gap between 2011 to 2017). Using a combo RT-q-PCR for detection of pan-Rabies and Pan-lyssavirus species, the authors were able to identify six samples containing vesiculoviruses or ledanteviruses positive results. After performing a combination of NGS sequencing, RACE and Sanger sequencing on gaps, six viral genomes were assembled and phylogenetically classified into three putative new viral species. The authors tentatively named the viruses as bat virus (YSBV), Taiyi bat virus (TYBV) and Qiongzhong bat virus (QZBV). All attempts to isolate the viruses from the tissues, both in vitro and in vivo, were not successful.
Mayor comment:
1. Results (page 6, line 255-271): All this data get lost in the text. I would suggest presenting this as a table that summarizes all these numbers, and replace figure 2, that in my opinion is not relevant (see mayor comment 2.). All this information gets lost in the text and it is hard to follow and compare.
2. Results (page 7, Figure 2): Figure 2 is not informative. This info could be moved to a table (see Mayor comments 1.). As now, a pie chart is taking a lot of space and does not have anything to the manuscript.
3. Results 3 (page 7, 3.2 Bat sample screening and description of positive specimens): It is key for the manuscript to establish what is your limit of detection. The authors touch this topic at the discussion (page 14-15, line 519-526) but have a sense on how this combo assay is performing becomes crucial to understand the success rate of detection. A simple experiment where a different known amount of a control virus (spike) on the matrix can be performed to have a sense on the limit of detection. In addition, the assay is limited by how conserve is the polymerase gene target region within all the diversity of this viral family. The authors should discuss this also. This will not take away that the team was successfully able to identify six viral sequences and three novel viruses.
4. Results (page 7, line 289-296): I am not sure what is the value of the “prevalence” data, considering the low number of positives found. For example, how would 1/9 (line 291) change if you had 19 samples? Would be 1/19 (5%) because and you “got lucky” finding that positive, or 3/19 (15%). I do not see much meaning to those numbers, as your total number of samples (even when impressive because it is a lot of work to obtain these numbers on the field) might be too small to give a statistical significance.
5. Results (table 2); table 2 is puzzling. The number cleaned reads (in the millions in all samples|) are dropping to a few thousand or less than hundreds when mapped. Why is that? Is 99.9% of your reads are host? Why do you see such a drop? I do not understand how you could build the genomes on Figure 3 with 22 mapped reads (for virus 1017) or even 0% mapped reads (virus D170190). The D170190 virus is puzzling. You got 0% mapped reads but you were still able to build an almost complete genome. This needs to be explained better. I am sincerely lost here.
6. Discussion (page 14, line 512-518): The authors state that “This result was not surprising because most of the samples were collected on apparently healthy and flying bats with nets when the left their roost caves at dusk, or directly in the colonies with landing nets” and after that, they state: “At the opposite, all new bat lyssavirus species have been detected in sick or dead animals, and the proportion of positive samples has been demonstrated to be higher when targeting this category of animals within bat colonies”. As far as I know, the bat’s health status is independent of the presence or absence of viral load, unless the virus is a detriment for the bat health. The authors state in the introduction (correctly) that most of the bats carry these viruses asymptomatically. The possibility to capture a viremic bat would depend on the virus dynamic and viral load at the moment of the capture unless the authors have evidence that these viruses make the bat sick and reduce their ability to fly and search for food.
7. Discussion (page 16, line 589-594): In the sentences: “Most interestingly and despite the close distance between the two caves, 21 amino acid mutations were found between YSBV isolates from Taiyi and Yinshui caves, which were suggestive of the presence of two different virus populations. Such markers would represent potential indicators to investigate the circulation of both YSBV isolates and associated bats (here Rhinolophus sinicus) between the two caves, although this hypothesis needs to be confirmed on a larger number of isolates.” Are the authors suggesting there is no bat circulation between the two caves (21 miles away)? Bats are famous for traveling and migration between caves. Maybe this is not true with this bat specie. I found remarkable these two viral populations (and accordingly the bat colonies in both caves) to be kept apart and isolated one from the other. Please, clarify.
Minor comments:
1. Introduction (page 2, line 50): Marburg virus (by Dr, Towner work) and other filoviruses (Lloviu virus, Mengla virus and Bombali virus) were detected and complete/almost complete genomes recovered from all of them. However, Ebola virus association with bats is more lose and controversial. There is serological data (Eidolon helvum, Epomophorus gambianus, Lissonycteris angolensis, Micropteropus pusillus, Mops condylurus, Rousettus aegyptiacus, Epomops franqueti, Hypsignathus monstrosus, and Myonycteris torquata).) However, since cross-reactivity of antibodies cannot be excluded, the detection of Ebola virus-specific antibodies could also indicate a related but perhaps not yet discovered virus strain. Three bat species tested positive by PCR methods (Epomops franqueti, Hypsignathus monstrosus and Myonycteris torquata), but no complete genome or viral isolation was ever successful from bats. The consensus is that Bats might be the reservoir of the Ebola virus, but that was not 100% established yet. As today, bats are only the most likely putative reservoir species of Ebola virus.
2. Introduction (page 2, line 76-77): The sentence is lacking reference(s).
3. Introduction (page 2, line 81-82): Please correct the sentence: “To date, two lyssaviruses isolated in Jilin and Taiwan have been reported recently”. To date and recently do not go well together.
4. M&M (page 3, line 123): in 2.4. Identification of bat species, the authors state that “Bat species were identified after partial sequencing of the cytochrome b (CYTB) gene”, however in line 119, the authors described that “For saliva swab samples, pools of 5 samples (40 μL 119 each) were similarly extracted”. How the species were determined in pooled samples? Please, clarify this step.
5. M&M (page 3 and 4) 2.5. Combo RT-q-PCR for Lysavirus screening. Were the positive controls (RABV CVS strain and Europe bat lyssavirus 1 8919FRA strain assayed in the same matrix as the samples (homogenized brain and saliva)? This would impact the assay performance (see Mayor comment 3.) and the interpretation of 2.6. Confirmation of doubtful samples by Sanger sequencing.
6. M&M (page 4, line 153-154): depletion of rRNA and mitochondrial RNA can be challenging in mammal cells. Most of the commercial kits were tested in either human or mouse samples. How does this kit perform in bats biospecimens? What the kit tested to address possible viral sequences lost?
7. M&M (page 5, line 217): Is there any rational reason why the primer sequences and PCR conditions would only be made available on request?
8. Figure 4 (page 11): The schematics showing the different species (bats, humans, NHP, rodent, bovine, etc.) does not belong to figure 4 but figure 5. Figure 4 only has the bat silhouette. Please, correct.
9. Discussion (page 15, line 520): the authors state: “we would have expected to identify at least 1 to 3 positive saliva samples from our dataset”. Why is that? With no context, this affirmation is vague and arbitrary. Please, explain.
10. Discussion (page 15, line 524-526): In the sentence: “Usually, similar studies are conducted on saliva samples collected in a smaller volume (nearly 200 μL of VTM, TRIzol or other), which is used full for RNA extraction”, what “used full for RNA extraction” means? Can this be replaced for: “and RNA was extracted from the entire volume”?
11. Discussion (page 15, line 542): In the sentence “…when compared to the most closely related other vesiculoviruses (JHBV) (Table S5).”, please delete the word “other”.
12. Discussion (page 16, line 616-620): Please try to avoid the use of adjectives like “massive”. How do you quantify “massive” in “Additional molecular screening and virus quantification in other available tissues revealed that the infection was massive in several organs, such as intestine, lung liver or heart, without any specific viral tropism”. Do you mean the virus was detected several organs? The same for page 17, line 624: “…symptoms (although the infection was already massive and systemic).”
13. Discussion (page 16, line 620-621): I believe there is a typo on: “This systemic infection raises questions about the impact of infected bats…” Should say …impact on infected bats…?
Author Response
Reviewer 2
Luo et al manuscript, entitled: “Characterization of novel rhabdoviruses in Chinese bats”, describes a molecular retrospective study for detection of lyssavirus in biospecimens collected from bats of Chinese origin. The team investigated 1044 brain and 3562 saliva samples from 63 different bats species. The sample collection span from 2006 to 2018 (with a gap between 2011 to 2017). Using a combo RT-q-PCR for detection of pan-Rabies and Pan-lyssavirus species, the authors were able to identify six samples containing vesiculoviruses or ledanteviruses positive results. After performing a combination of NGS sequencing, RACE and Sanger sequencing on gaps, six viral genomes were assembled and phylogenetically classified into three putative new viral species. The authors tentatively named the viruses as bat virus (YSBV), Taiyi bat virus (TYBV) and Qiongzhong bat virus (QZBV). All attempts to isolate the viruses from the tissues, both in vitro and in vivo, were not successful.
Mayor comment:
- Results (page 6, line 255-271): All this data get lost in the text. I would suggest presenting this as a table that summarizes all these numbers, and replace figure 2, that in my opinion is not relevant (see mayor comment 2.). All this information gets lost in the text and it is hard to follow and compare.
As suggested by Reviewer 2, we replace Figure 2 with Table 1 (see also Major comment 2). We completed Figure 1 (which includes now the number of samples per province) and we modified this results section for clarification (lines 270-281).
- Results (page 7, Figure 2): Figure 2 is not informative. This info could be moved to a table (see Mayor comments 1.). As now, a pie chart is taking a lot of space and does not have anything to the manuscript.
According to the recommendation of Reviewer 2, we replaced Figure 2 with Table 1 (line 282).
- Results 3 (page 7, 3.2 Bat sample screening and description of positive specimens): It is key for the manuscript to establish what is your limit of detection. The authors touch this topic at the discussion (page 14-15, line 519-526) but have a sense on how this combo assay is performing becomes crucial to understand the success rate of detection. A simple experiment where a different known amount of a control virus (spike) on the matrix can be performed to have a sense on the limit of detection. In addition, the assay is limited by how conserve is the polymerase gene target region within all the diversity of this viral family. The authors should discuss this also. This will not take away that the team was successfully able to identify six viral sequences and three novel viruses.
We appreciate and thank the Reviewer 2 for this important comment. As an example, we selected one representative (isolate D170001) of the Yinshui bat virus for the determination of the limit of detection with the SYBR Green RT-qPCR. We described this determination in the Materials and Methods section (lines 146-160) as well as in the Results section (lines 303-309) with the supplementary figure S3. For this determination, we used water instead of more relevant matrix such as saliva or brain due to lack of availability of such samples. However, even with nuclease free water, we demonstrated that the limit of detection was high with the isolate D170001 (1x107 copies per assay, Figure S3). This low sensitivity was expected because the SYBR Green RT-qPCR was initially designed for lyssavirus detection (see Dacheux et al., 2016), which was the initial goal of your study.
We also presented in the supplementary figure S2 the nucleotide sequence alignment of the regions targeted by the primers of the pan-lyssa RT-qPCR for the 161 rhabdovirus sequences (including the 6 Chinese bat rhabdoviruses) used for the phylogenetic tree in Figure 3. This Figure S2 give a detailed overview of the genetic diversity in this relatively conserved region of the polymerase gene.
Lastly, we also discussed this element in the Discussion section, lines 577-580: “Despite this level of conservation, the presence of potential mutations in these regions, depending on the rhabdovirus species considered, could negatively impact the sensitivity of this technique, and lead to not detecting animals with low viral loads (Figure S2).”
We did not test the probe-based (TaqMan) pan-RABV RT-qPCR (mainly designed to the detection for rabies virus isolates) because this system failed to detect any rhabdovirus in our study.
We also included now some additional data (Table S9) corresponding to the screening by the specific RT-PCRs of the samples collected from Rhinolophus sinicus and affinis originated from the same locations of the positive specimens (paragraph “3.6. Bat sample screening by specific RT-qPCRs” in the Results section, lines 499-505). We confirmed the previous results obtained on the same samples with the pan-lyssa RT-qPCR, and we tested additional ones, which were found all negative.
- Results (page 7, line 289-296): I am not sure what is the value of the “prevalence” data, considering the low number of positives found. For example, how would 1/9 (line 291) change if you had 19 samples? Would be 1/19 (5%) because and you “got lucky” finding that positive, or 3/19 (15%). I do not see much meaning to those numbers, as your total number of samples (even when impressive because it is a lot of work to obtain these numbers on the field) might be too small to give a statistical significance.
We agree with the Reviewer 2 and we modified “prevalence” with “detection rate”. In addition, we added the following limitation lines 324-325: “However, the number of positive samples or total samples tested remained low.”
- Results (table 2); table 2 is puzzling. The number cleaned reads (in the millions in all samples|) are dropping to a few thousand or less than hundreds when mapped. Why is that? Is 99.9% of your reads are host? Why do you see such a drop? I do not understand how you could build the genomes on Figure 3 with 22 mapped reads (for virus 1017) or even 0% mapped reads (virus D170190). The D170190 virus is puzzling. You got 0% mapped reads but you were still able to build an almost complete genome. This needs to be explained better. I am sincerely lost here.
We apologize for the confusion causes by this Table 2. We deleted this figure to avoid such confusion. In order to clarify the situation for Reviewer 2, we initially performed NGS sequencing on the positive samples, then completed the genome sequence determination by Sanger sequencing, especially for the low or very low coverage rate (even no coverage) of some samples (such as D170190, 1017 or 1127 isolates). The last mapping step was informative for verification with this approach only for high covered samples such as D170001 or D170022.
- Discussion (page 14, line 512-518): The authors state that “This result was not surprising because most of the samples were collected on apparently healthy and flying bats with nets when the left their roost caves at dusk, or directly in the colonies with landing nets” and after that, they state: “At the opposite, all new bat lyssavirus species have been detected in sick or dead animals, and the proportion of positive samples has been demonstrated to be higher when targeting this category of animals within bat colonies”. As far as I know, the bat’s health status is independent of the presence or absence of viral load, unless the virus is a detriment for the bat health. The authors state in the introduction (correctly) that most of the bats carry these viruses asymptomatically. The possibility to capture a viremic bat would depend on the virus dynamic and viral load at the moment of the capture unless the authors have evidence that these viruses make the bat sick and reduce their ability to fly and search for food.
We thank the Reviewer 2 for this important comment. As far as it concerns rabies virus, and lyssaviruses in general, these viruses can induce disease (clinical sign of rabies, including abnormal behavior and disability to fight) and death in bats. Multiple evidences were demonstrated in this animal reservoir with EBLV-1, EBLV-2, BBLV, RABV, ABLV, etc, even if we can not exclude that some bats can survive to the infection (see for example studies on EBLV-1 and Myotis myotis in Spain, Amengual et al. 2007) or/and can remain asymptomatic.
For instance, most if not all positive cases in bats in Europe were detection via a passive surveillance based on the diagnosis on dead, symptomatic bats or on animals presenting abnormal behavior. Thus, as indicated in our manuscript, targeting symptomatic animals could increase the rate of positive detection for lyssavirus, as exemplified with the study on GBLV in Sri-Lanka (Gunawardena et al. 2016), where the authors reached a high positive rate (6.4%, 4/62) when targeting grounded bats.
For clarification, we modified this part of the discussion lines 546-549: “Indeed, multiple evidences demonstrated that bats are sensitive to lyssavirus infection, which can induce signs of rabies (including disability to flight, abnormal behavior, aggressiveness) and ultimately lead to death [25,66-68], even if we cannot exclude asymptomatic infection”.
- Discussion (page 16, line 589-594): In the sentences: “Most interestingly and despite the close distance between the two caves, 21 amino acid mutations were found between YSBV isolates from Taiyi and Yinshui caves, which were suggestive of the presence of two different virus populations. Such markers would represent potential indicators to investigate the circulation of both YSBV isolates and associated bats (here Rhinolophus sinicus) between the two caves, although this hypothesis needs to be confirmed on a larger number of isolates.” Are the authors suggesting there is no bat circulation between the two caves (21 miles away)? Bats are famous for traveling and migration between caves. Maybe this is not true with this bat specie. I found remarkable these two viral populations (and accordingly the bat colonies in both caves) to be kept apart and isolated one from the other. Please, clarify.
We totally agree with the Reviewer 2 regarding this interesting and puzzlingly result. Unfortunately, little is known about the ecological factors of Rhinolophus sinicus and more especially about their circulation and migration. We have modified this part of the discussion accordingly.
Minor comments:
- Introduction (page 2, line 50): Marburg virus (by Dr, Towner work) and other filoviruses (Lloviu virus, Mengla virus and Bombali virus) were detected and complete/almost complete genomes recovered from all of them. However, Ebola virus association with bats is more lose and controversial. There is serological data (Eidolon helvum, Epomophorus gambianus, Lissonycteris angolensis, Micropteropus pusillus, Mops condylurus, Rousettus aegyptiacus, Epomops franqueti, Hypsignathus monstrosus, and Myonycteris torquata).) However, since cross-reactivity of antibodies cannot be excluded, the detection of Ebola virus-specific antibodies could also indicate a related but perhaps not yet discovered virus strain. Three bat species tested positive by PCR methods (Epomops franqueti, Hypsignathus monstrosusand Myonycteris torquata), but no complete genome or viral isolation was ever successful from bats. The consensus is that Bats might be the reservoir of the Ebola virus, but that was not 100% established yet. As today, bats are only the most likely putative reservoir species of Ebola virus.
We thank Reviewer 2 for this important and helpful reminder. We included “or suspected” line 55.
- Introduction (page 2, line 76-77): The sentence is lacking reference(s).
We added the references “[35,36]” line 79.
- Introduction (page 2, line 81-82): Please correct the sentence: “To date, two lyssaviruses isolated in Jilin and Taiwan have been reported recently”. To date and recently do not go well together.
Correction done.
- M&M (page 3, line 123): in 2.4. Identification of bat species, the authors state that “Bat species were identified after partial sequencing of the cytochrome b (CYTB) gene”, however in line 119, the authors described that “For saliva swab samples, pools of 5 samples (40 μL 119 each) were similarly extracted”. How the species were determined in pooled samples? Please, clarify this step.
We apologize for this lack of clarity. We modified as follow: “Bat species was determined on morphological features, and doubtful or unidentified species were identified after partial sequencing of the cytochrome b (CYTB) gene.”
- M&M (page 3 and 4) 2.5. Combo RT-q-PCR for Lysavirus screening. Were the positive controls (RABV CVS strain and Europe bat lyssavirus 1 8919FRA strain assayed in the same matrix as the samples (homogenized brain and saliva)? This would impact the assay performance (see Mayor comment 3.) and the interpretation of 2.6. Confirmation of doubtful samples by Sanger sequencing.
The validation of the combo RT-q-PCR for lyssavirus diagnostic was done in Dacheux et al., 2016 (DOI:10.1371/journal.pntd.0004812). For this validation process, we used plasmids encoded the target sequence and diluted in nuclease free water, as well as RNA extracted from titrated virus suspension and spiked into RNA extracted from brain. Several different lyssavirus species were tested, included RABV CVS strain and EBLV1 strain 8918FRA. In addition, this combo RT-q-PCR was already evaluated for the rabies diagnosis in human, using 120 saliva and 12 brain samples.
- M&M (page 4, line 153-154): depletion of rRNA and mitochondrial RNA can be challenging in mammal cells. Most of the commercial kits were tested in either human or mouse samples. How does this kit perform in bats biospecimens? What the kit tested to address possible viral sequences lost?
Reviewer 2 is totally right regarding the challenge of rRNA depletion in other species (and especially bats) that the classical ones (i.e. human, rat, mouse). Unlike the kits using specific nucleotide probes directed against the sequences of ribosomal acids (such as kits from NEB with NEBNext rRNA Depletion Kit or from Illumina with Ribo-Zero Gold rRNA Removal Kit or with Ribo-Zero Gold rRNA Removal Kit), the protocol used was based on the Terminator 5´-Phosphate-Dependent Exonuclease which processively digests RNA with 5´-monophosphate ends but not RNAs with 5´-triphosphate, 5´-cap or 5´-hydroxyl groups starting from the 5´ end, independently of species origin. Evaluation of this depletion protocol for lyssavirus sequencing has been done by Martson et al., 2013 (https://doi.org/10.1186/1471-2164-14-444). This step is now included in our own NGS protocol, whatever the species origin of the sample sequenced. We did not perform an evaluation of its performance on bat specimens.
- M&M (page 5, line 217): Is there any rational reason why the primer sequences and PCR conditions would only be made available on request?
Requested information was added to the manuscript.
- Figure 4 (page 11): The schematics showing the different species (bats, humans, NHP, rodent, bovine, etc.) does not belong to figure 4 but figure 5. Figure 4 only has the bat silhouette. Please, correct.
We thank the Reviewer 2 for identifying this error. Correction has been done.
- Discussion (page 15, line 520): the authors state: “we would have expected to identify at least 1 to 3 positive saliva samples from our dataset”. Why is that? With no context, this affirmation is vague and arbitrary. Please, explain.
This estimation of the number of positive samples was based on a positive rate of 0,1%, as indicated in the discussion “previous studies demonstrated that the positive rate for lyssavirus detection was expected to be very low, around 0,1% in brain or saliva sample [71–73].” lines 552-553). Thus, if 3500 samples are tested, we can expect to identify at least 3 positive samples. For clarification, we modified the sentence as follow: “However, although the number of brains tested (nearly 1,000 samples) may explain the lack of positivity, we would have expected to identify around 3 positive saliva samples from our dataset (nearly 3,500 samples), based on this positive detection rate of 0,1%.” lines 554-556.
- Discussion (page 15, line 524-526): In the sentence: “Usually, similar studies are conducted on saliva samples collected in a smaller volume (nearly 200 μL of VTM, TRIzol or other), which is used full for RNA extraction”, what “used full for RNA extraction” means? Can this be replaced for: “and RNA was extracted from the entire volume”?
Modified accordingly.
- Discussion (page 15, line 542): In the sentence “…when compared to the most closely related other vesiculoviruses (JHBV) (Table S5).”, please delete the word “other”.
Correction done.
- Discussion (page 16, line 616-620): Please try to avoid the use of adjectives like “massive”. How do you quantify “massive” in “Additional molecular screening and virus quantification in other available tissues revealed that the infection was massive in several organs, such as intestine, lung liver or heart, without any specific viral tropism”. Do you mean the virus was detected several organs? The same for page 17, line 624: “…symptoms (although the infection was already massive and systemic).”
We modified the different part of the manuscript accordingly to the Reviewer 2: “These results demonstrated that multiple organs can be infected, with a high viral load.” lines 490-491, and Additional molecular screening and virus quantification in other available tissues revealed that the infection was multi-organs, such as intestine, lung liver or heart, associated to a high viral load lines 667-668.
- Discussion (page 16, line 620-621): I believe there is a typo on: “This systemic infection raises questions about the impact of infected bats…” Should say …impact on infected bats…?
Correction done.
Round 2
Reviewer 2 Report
The authors have responded and addressed most of my concerns.